# Designing tuberculosis elimination framework through participatory processes in Ethiopia: Results from stakeholders' discussions

Mulatu Biru[1], Daniel G. Datiko[2], Degu Jerene[3], Asfawesen G. Yohannes[1], Yohannes Molla[2], Pedro Suarez[4], Wondmiu Gebrekiros[5], Anteneh Kassa[5], Yewulsew Kassie[5], Zewdu G. Dememew[2]*

1 USAID Eliminate TB Project, KNCV Tuberculosis Foundation, Addis Ababa, Ethiopia, 2 USAID Eliminate TB Project, Management Sciences for Health (MSH), Addis Ababa, Ethiopia, 3 KNCV Tuberculosis Foundation, The Hague, Netherlands, 4 MSH, Arlington, Virginia, United States of America, 5 USAID Ethiopia, Development Program Specialist, Infectious Diseases Cluster, Addis Ababa, Ethiopia

* zgahsu@msh.org

## Abstract

### Background

Tuberculosis elimination requires implementation of multi-pronged, complex interventions, but there is limited understanding of frameworks for guiding implementation. Our aim was to explore the feasibility of applying multipronged package of TB elimination interventions guided using the Medical Research Council (MRC) framework.

### Methods

This study is a component of a larger quasi-experimental study aimed at demonstrating what TB elimination interventions would entail under routine TB program conditions titled *Demonstrating Multipronged and Optimized Novel Strategies to Reinforce Actions Targeted at Eliminating Tuberculosis.* During its preparatory phase in February 2023, we employed a participatory action research design which involved gathering feedback and insights from TB program managers, healthcare workers and community representatives about the study interventions. The study participants were selected from three districts of Amhara, Oromia, and Southern Nations, Nationalities and Peoples Regions of Ethiopia. Two senior researchers took detailed notes during the meetings using topic guided questions. Each topic guided question was thoroughly explored until no new issues emerged. Thematic analysis was conducted to summarize the reflections of the study participants from the three sessions conducted in the three regions.

### Results

108 participants attended the preparatory phase discussions which were summarized under five sub-themes under the main theme of "*Looking for local evidence compulsory for TB elimination.*" These included; (1) the importance of community engagement which

**Data availability statement:** All relevant data are within the manuscript and its Supporting Information files.

**Funding:** The author(s) received no specific funding for this work.

describes the need to include health extension workers, health development army, and other community structures to support the study, (2) government interest toward local evidence, which describes how local evidence is essential to support the TB program and address challenges, (3) homogeneity of officials' commitment across the regions which is reflected in their immense support to undertake the study and suggested working together for the study follow-up,(4) cross-cutting issues or multiple factors such as nutrition, sociocultural factors, livelihood, and housing, were considered, and (5) the importance of establishing TB treatment supporter, which describes the critical role that TB treatment supporters play in ensuring successful treatment outcomes.

## Conclusions

This study highlights the importance of MRC tailored protocol development and feasibility in effective implementation. Key lessons include the need for early stakeholder engagement, streamlined communication, and proactive risk management. Strong community engagement, government commitment, and addressing social determinants were critical to TB elimination. These findings emphasize the value of a collaborative, multi-pronged approach during the adoption and implementation of the TB elimination framework.

## Introduction

Tuberculosis (TB) continues to be one of the leading causes of death worldwide where the African region accounted for nearly one-fourth (23%) of the estimated global burden for 2022 [1]. The African region is one of the WHO regions that has achieved an estimated TB incidence reductions of more than 20% since 2015, thus surpassing the first milestone of the End TB Strategy for 2020. This progress could be realized through universal access for diagnosis and treatment, addressing TB determinants such as undernutrition, HIV infection, alcohol use disorders, smoking, diabetes, poverty, and other cross-cutting factors include individuals residing in slums, pastoralist communities, indoor air pollution and changes in the absolute level of GDP per capita. Additionally, more than 85% of TB cases could be treated successfully by prioritizing community engagement, referring individuals with TB symptoms to health facilities, providing treatment support and fostering a culture of robust collaboration across multiple sectors, and implementing accountability mechanisms [1].

Intensified research and innovation are among the three pillars of the global END TB strategy toward achieving the global target of ending TB by 2035 [2,3]. To foster greater research and innovation, this should be augmented by investments in research and development by promoting successful collaboration among stakeholders which involves donors, scientific experts, TB program managers, engagement of the community and civil society organizations working together effectively. By encouraging these intersectoral and multisectoral partnerships, an environment that empowers intensified research and innovation efforts can be created [4].

Ethiopia is in the list of high burden countries of TB and TB/HIV with high rates of TB incidence of 126 per 100,000 population and TB related mortality of 17 per 100,000 population in 2022. However, there has been significant reduction in Ethiopia's TB burden. TB incidence (per 100,000) declined from 277 in 2011 to 192 in 2015, and then to the current level of 126 in 2022, a decline by 34.4%. Similarly, the TB mortality rate declined from 24 in 2015 to 17 in 2022, a reduction by 29.2%. These were achieved because of intensified and targeted interventions undertaken by the national TB program [1,2,5]. However, the impact

of Internally Displaced Persons (IDPs), lack of awareness about TB, stigma, and malnutrition could cause the worst impact on rural communities in Ethiopia. In addition, rapid urban migrations often result in overcrowded cities, giving rise to slum towns with inadequate and poorly ventilated housing. These deficiencies pose significant health risks to residents and can serve as potential transmission grounds for infectious diseases, mainly TB. Approximately 62% of the urban population in sub-Saharan Africa resided in shanty towns [6]. Similarly, urbanization and population growth in Ethiopia is expected to further increase TB cases in the country [7]. Therefore, it is essential to be ambitious and define TB elimination packages in the country to deal with all key drivers of TB incidence in the country. Although there are some experiences of TB elimination trials [8–11], most evidence is concentrated in developed countries with low TB incidence.

It is therefore important to generate local evidence from low-income but high TB burden settings such as Ethiopia [12–14]. This will enable informed decision-making and promote a better understanding of the specific context and needs of these countries. The Demonstrating Multipronged and Optimized Novel Strategies to Reinforce Actions Targeted at Eliminating Tuberculosis (DeMONSTRATE TB) study in Ethiopia aims at generating local evidence toward preparing a framework for TB elimination in low-income countries with high TB burden. In this paper, we explore the feasibility of implementing a proposed package of interventions to eliminate tuberculosis in Ethiopia and beyond based on the experiences from the preparatory phase of DeMONSTRATE-TB.

## Methods

### The conceptual framework

Because of the complex and comprehensive nature of the intervention package, we used the Medical Research Council (MRC) model of complex interventions as the methodological basis [15] which includes development, feasibility, evaluation, and implementation phases (Fig 1). Implementation research design based on the MRC Framework for complex interventions is a rigorous approach that can help to ensure that interventions are effective, efficient, and sustainable. The complexity of an intervention in the context of this study can be attributed to several factors related to the intervention itself [15].

**Study design.** This is a participatory action research (PAR), a component of the larger DeMONSTRATE TB study, which aims to showcase the effectiveness and feasibility of implementing an enhanced community-wide TB Preventive Treatment (TPT) alongside intensified active TB case finding, enhanced diagnostic capabilities, and increased awareness efforts. The goal is to generate evidence that will contribute to the development of a comprehensive framework for TB elimination that can be applied on a wider scale in high burden and resource constrained settings. This PAR study, therefore, will provide evidence regarding the feasibility of implementing the proposed study design, including opportunities and challenges. We deployed a PAR design [16] that involved in-depth conversations with mid-level TB program managers, healthcare workers and community representatives from different regions aiming at assessing the viability of the study and refine the operational procedures during the study launching, and orientation ensuring that the study is executed smoothly and effectively.

Participatory research is a collaborative approach to explore the active involvement of stakeholders, community members, or people with lived experience in the research process. This approach recognizes that the people who are most affected by the research are the ones who are best placed to identify research priorities and provide valuable insights and perspectives. In participatory research, researchers work in partnership with these people to co-create

**Fig 1. Theory-informed design based on the MRC framework for the complex interventions targeted at TB elimination.** It has four steps: 1) Development phase where proposal is developed after experts' discussions and brainstormings; 2) Feasibility or piloting phase where the study tools are tested and piloted; 3) Evaluation steps will assess the effectiveness and changing process of the interventions, and 4) Implementation phase where the study findings are disseminated.

knowledge that is relevant, meaningful, and usable. By working together, researchers and stakeholders can produce research that is more inclusive, culturally sensitive, and impactful by engaging communities in large-scale interventions [17,18].

**Study setting.** The study was started in three different regions of Ethiopia, namely Amhara, Oromia, and South Ethiopia Regional States. The TB case notification rates (CNR) data for the years 2016 to 2021 was gathered from the annual reports of each woreda in the three regions. The study and control woredas (lower-level administrative districts) were matched and selected based on high TB CNR. In collaboration with the Ministry of Health and regional health bureaus, the USAID Eliminate TB Project identified the study woredas with active involvement from zonal, woreda, and community health extension workers (HEWs) leaders in the site selection process.

Woredas with an average TB CNR of more than 140 cases per 100,000 population were selected, one woreda from each region. For each intervention woreda, we selected a control woreda from a non-adjacent area within the same region using the same criteria. Prior to piloting the study, extensive discussions and site visits were conducted with regional, zonal, and woreda health offices in each region (Fig 2, S1 Table). These discussions focused on the implementation process of the TB elimination study, mapping the study sites, and identifying control woredas. TB elimination as a public health problem is defined as the achievement of an incidence rate of less than 1 case of infectious TB per million population [19].

**Participants and procedure.** We selected participants from the DeMONSTRATE-TB study woredas of Amhara, Oromia, and South Ethiopia regions.

To ensure a diverse range of perspectives in this study, the participants were identified from the government health system and community by involving regional, zonal, and woreda health system managers. These individuals play key roles in leadership, monitoring, and

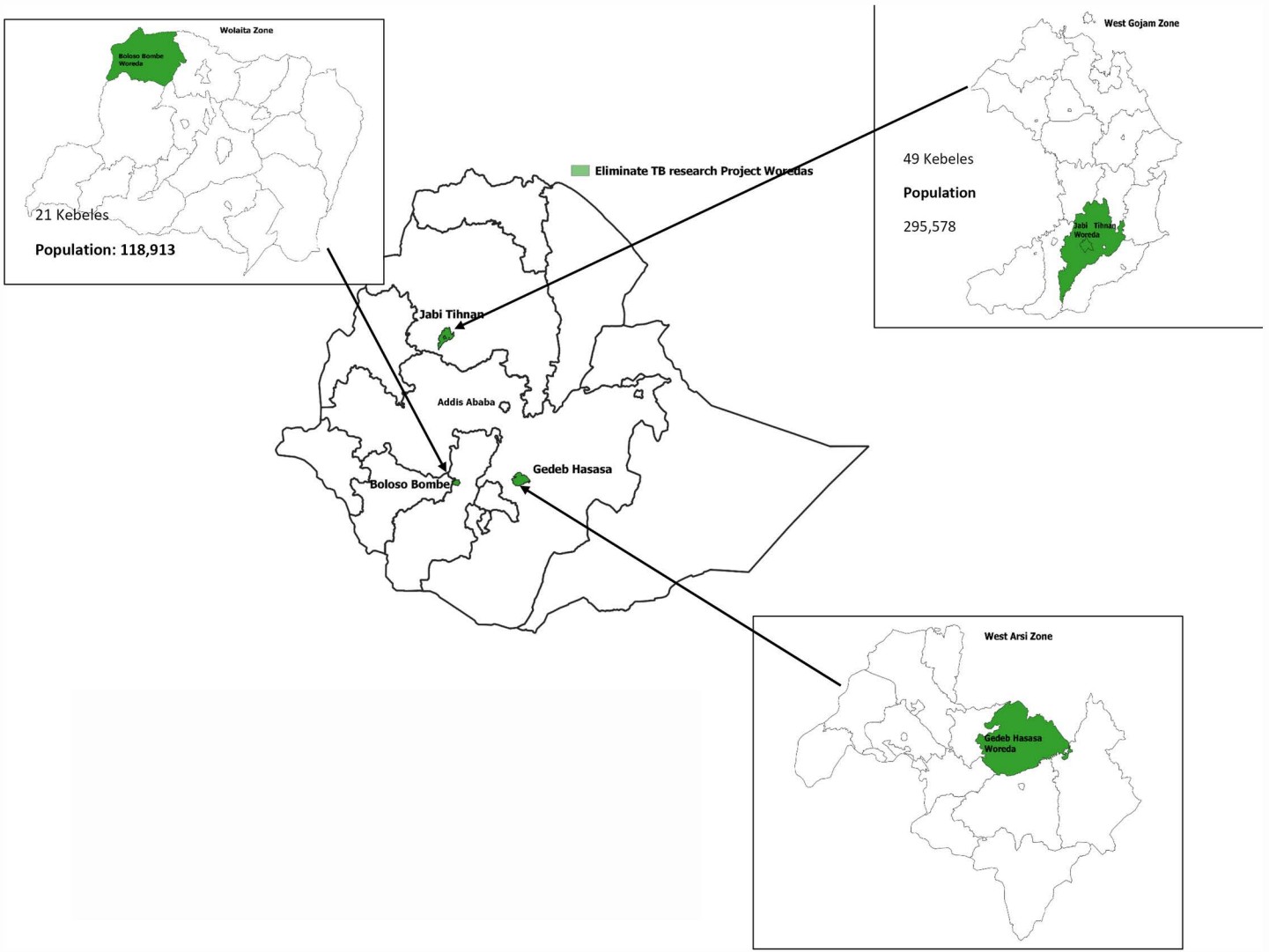

**Fig 2. Indicates the three study districts.** 1) Bolosso Bombe in Southern Ethiopia with a population of 120k, 31k households, four health facilities and 21 kebeles (smaller administrative unit), 2) Gedeb Hasasa in Oromia with a population of 290K, 71k households, eight health facilities and 25 kebeles and 3) Jabi Tihnan in Amhara region with a population of 295k, 75k households, ten health facilities and 49 kebeles.

decision-making regarding TB program implementation, policy changes, disease prevention, and control. The participants were selected from different levels of the health system, including disease prevention and control heads, TB program focal persons from regional health bureau, zonal health departments, district/woreda health offices, health facilities, and community HEWs team leaders. Throughout the development stage, we considered the overall experience and feasibility of the study, as reflected during the kick-off and study orientation. We strongly believe that input from a diverse range of individuals with varied expertise would provide valuable insights for planning and implementing the study, which aims to generate evidence supporting the development of a TB elimination framework.

**Data collection.** Data collection was conducted between February 20 and March 01, 2023. The discussions were held with the identified participants to ensure that we had relevant insights from key stakeholders. Two senior researchers, experienced in qualitative research

have been tasked with documenting these reflections to ensure that we capture all the valuable insights shared during the discussions. The data collection was performed during the three day long launching and orientations in three regions. We organized participatory discussions for each region to gather information. Additionally, we had one-on-one discussions with TB experts to delve deeper into area-specific details. Considering the contextual differences, we treated saturation separately for each region. For cross-cutting information, however, discussions in all the three districts were used to ensure saturation. We employed triangulation by incorporating multiple perspectives and viewpoints during the data collection process. We sought input from different stakeholders and experts with varying levels of responsibilities and gender, as this was believed to capture diverse insights related to the study topic. This rigorous process, in turn, enhanced the credibility of the findings and ensured saturation in the information obtained. Finally, we consolidated all the information for the analysis (S2 Table).

**Data analysis.** Thematic analysis was carried out to summarize the reflections from the study participants. The researchers took a rigorous approach to the analysis by having two senior researchers read and re-read the summary of reflections over the course of three different day-long study launchings, across various contexts and participants from three different regions. This allowed them to immerse themselves in the content and gain a deeper understanding of the data.

After this process, the researchers developed a codebook, which is a document that outlines the themes and subthemes identified in the data. The codebook served as a guide for the researchers to systematically analyze the data and identify common patterns or themes. We utilized the Open Code 4.03 tool for coding the transcribed qualitative data.

The use of thematic analysis allowed the researchers to identify important themes and patterns in the data, which can inform future research and practice in the field. Additionally, the rigorous approach taken by the researchers in developing the codebook and analyzing the data enhances the credibility and trustworthiness of the findings.

## Ethical considerations

We sought and secured ethical clearance to undertake the study from Population Services International (PSI) on December 16, 2022, and modification request approval on May 31, 2023, (REB# 37.2022) and Armauer Hansen Research Institutes (AHRI) on November 15, 2022 (PO-44-22). Oral consent was requested and obtained prior to the brainstorming and interview. The consent process was documented by the research team through written records indicating the participant's agreement, which were signed by both the interviewer and a witness present during the consent process.

## Findings

The results were described based on the first two phases of the MRC framework for complex interventions in healthcare including protocol development and piloting the feasibility of the study. Of the total 210 participants, 108 participants attended the launching, and 102 health workers attended data collection training (Table 1, S3 Table). These were TB program managers, focal persons, community workers, and health care workers from the health facilities.

The key findings of this study were explained under two broader categories, which include "Experience from MRC tailored protocol development" and "Experience from MRC tailored feasibility and piloting of the study."

**Experience from MRC tailored protocol development.** USAID Elimination TB project in Ethiopia developed a research project which intended at generating evidence through the demonstration of multipronged interventions which are evidenced at eliminating tuberculosis

**Table 1. Summary of participants of DeMONSTRATE-TB study launching, Feb 20–Mar 01, 2023.**

| Study area | Type of event | Number of participants | | |
|---|---|---|---|---|
| | | Male | Female | Total |
| Wolaita Zone, Bolosso Bombe | Study Launching | 20 | 1 | 21 |
| West Arsi Zone, Gedeb Hasasa | Study Launching | 26 | 7 | 33 |
| West Gojjam Zone, Jabi Tihnan, | Study Launching | 48 | 6 | 54 |
| Total | | 94 | 14 | 108 |

and supported by the global and national tuberculosis guidelines. However, applying this plan along the program implementation is not as simple as an initial project insight due to several reasons. The major reasons include but not limited to the experts engaged at various levels of the protocol development and not everyone has the same level of knowledge and experience about the research plan which made the protocol development process to be delayed.

On the other hand, the project design changed midway, January to April 2022, through the protocol development, which altered the project from its initial plan. The quantitative initial study design was a cluster randomized trial with kebele as a cluster unit. This has been shifted to quasi-experimental taking the whole woreda or districst as study unit. This was due to highly anticipated contamination of the intervention among kebeles in a district. Hence, the change has created a need for multiple revisions and iteractions between research experts and decision makers.

In addition, meeting ethical clearance requirements for clinical trials, both locally and internationally, poses significant challenges. These challenges encompass navigating complex regulatory environments, obtaining feedback from rigorous ethical review processes, addressing specific requirements, developing tailored informed consent and assent for diverse study participants, and addressing concerns related to data capturing, management, and sharing. These tasks required considerable time and effort.

This protracted endeavour, spanning a duration of seven months from July 2022 to January 2023, is primarily attributed to the diverse set of prerequisites mandated by different ethics review boards. As a result, this unforeseen hurdle has impeded the advancement of the research project and necessitated the postponement of significant milestones.

**Experience from MRC tailored feasibility and piloting.** The participatory discussions were articulated in five sub-themes under the main theme of "Looking for local evidence compulsory for TB elimination" (Fig 3). Sub-themes include (1) the importance of community engagement which describes the need to include HEWs, the health development army (HDA), and other community structures to support the study. (2) Government interest toward local evidence, which describes how local evidence is essential to support the TB program and address challenges. In (3) homogeneity of officials' commitment across the regions, they showed their immense support to undertake the study yet suggested working together for the study follow-up. In (4) considering multiple factors, issues, such as nutrition, sociocultural factors, livelihood, and housing, were considered. Finally, in (5) the importance of assigning TB treatment supporter, which describes the critical role that TB treatment supporters play in ensuring successful treatment outcomes both for the TB prevention therapy (TPT) and anti-TB medications at the community level.

## Looking for local evidence important for TB elimination

This overall theme is focused on the need for local evidence in the effort to eliminate TB. It highlights the importance of collecting data from local communities to better understand the

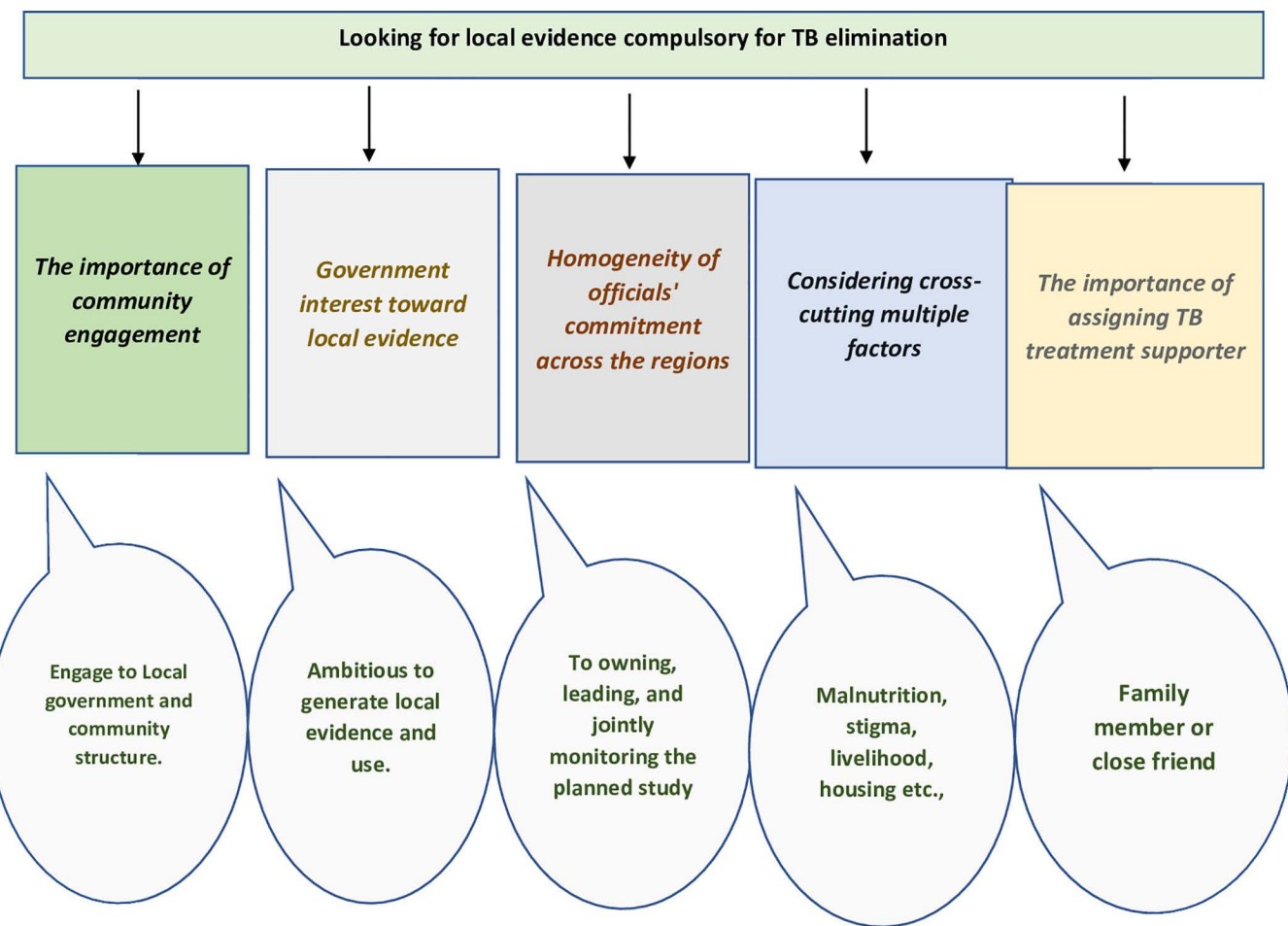

**Fig 3. Summary of the findings from the thematic analysis of the participatory stakeholders' discussions under the main theme of ''Looking for local evidence compulsory for TB elimination'.** There were five main sub-themes: 1) the importance of community engagement; 2) Government interest toward local evidence; 3) homogeneity of officials' commitment across the regions; 4) considering multiple factors; and 5) the importance of assigning TB treatment supporter.

specific challenges of TB in different contexts and to develop effective strategies for eliminating the disease. Most of the attendees stated that eliminating tuberculosis is impossible without having strong and contextually relevant local evidence. To achieve this, there is a need to have a unified commitment from health officials' local government, public and private sectors, healthcare providers, multidisciplinary experts, and active community engagement at large across regions. Overall, mid-level health managers and healthcare workers invited from the government health system applauded that this is a very high time for the health system to look for local evidence that could be used as a strategic solution for the ever-growing challenge of tuberculosis. By doing so, the system can take proactive measures to combat the spread of the disease and improve public health outcomes.

**The importance of community engagement.** During the interactive discussions, the participants underlined that any of the novel approaches and strategic directions should consider the engagement of existing community structures. There are important community structures including HEWs, the HDA, and other community structures such as religious leaders, kebele leaders, and other influential individuals in the community should be engaged to support health system strengthening practices and the study.

*"…Being involved from officials and the community at multiple levels is very important when it comes to conducting a study. This is because it not only helps to take ownership of the study but also ensures that proper procedural steps are followed before obtaining formal permission to proceed."* M-WAZ-001

Participants emphasized that community engagement is not optional but mandatory when conducting a TB-related study. It is crucial for the community to fully understand the study's objectives and potential outcomes to meaningfully contribute from the beginning. This inclusive approach is believed to foster ownership, collaboration, and ultimately improves the effectiveness and sustainability of the intervention.

*"Active community engagement and awareness empower them to contribute significantly at every stage of the intervention. By actively participating, the community becomes an integral part of the process, bringing their unique perspectives, knowledge, and resources. Woreda representative-M-WGZ-001.*

The participants in the study also mentioned that there were occasions when research projects were started without actively involving the community and regional government officials, resulting in their eventual termination. Therefore, it is crucial to clearly engage the community and regional government efforts right from the beginning when drafting the proposed protocol. By doing so, it paves the way to ensure effective collaboration and prevent any potential roadblocks that may arise later in the research process.

*"…Our experience has shown that studies which do not follow the necessary steps, such as obtaining approval from regional research approving bodies in addition to the national IRB, can be stopped."* M-WAZ-001

*"Active community engagement… from outset facilitates the smooth progress of the study. Failure to engage at least with community leaders may pose a challenge to the success of the study."* Woreda representative-M-WGZ-001

The study team learnt that engaging health care workers from the study district and the community facilitated and streamlined the implementation of the study. It further helps for the purpose of sustainability and ownership of the interventions.

**Government interest toward local evidence.**  According to the participants' reflection, TB elimination cannot be realized unless there is ongoing evidence generating mechanism which is framed based on the local solutions. Thus, there is a need to generate local evidence and test its effectiveness and practicability to support the TB program and address challenges. They firmly expressed their interest in supporting the collection and use of local evidence in the effort to eliminate TB. The participated TB experts explained that there are a lot of factors that could be considered for TB elimination and hence further evidence needed to be generated that could be considered in the national TB program strategic plan. They understand that government involvement is crucial for ensuring that resources are allocated appropriately and that policies are effective.

*"…Our ambition goes beyond simply contributing to a research project. This study is an intervention that holds the promise of revolutionizing the current practice of TB screening, diagnosis, and linking patients to treatment and care."* One of the heads of zonal disease prevention and control." *M-WAZ-002*

The regional and zonal TB focal persons were quite interested in the evidence generation. This could assist in the policy change and bring efficient interventions to deal with TB epidemic. Above all, it is stressed that local and context-based information are paramount important to improve care of TB cases.

*''…so far, we have been implementing the national guideline per the global recommendations. However, it could be beneficial if local information could be an input for nation, regional and districts TB integrations.'' M-WZ-003*

**Homogeneity of officials' commitment across the regions.**  The mid-level managers showed their immense support to undertake the study yet suggested working together for the study follow-up. Among the reflections, one interesting aspect to note is the demonstration of a package of interventions towards TB elimination. It has a clear indication of the government's ambition to not only end TB burden in the selected study areas but also to extend their efforts beyond those areas. This shows a strong commitment to achieving the goal of eliminating TB altogether.

*"…We are fully committed to ensuring the success of this study project. We believe that the package of interventions and resources offered by the project team is excellent, and we are thrilled to be able to support it. By working together and leveraging our collective strengths, we are confident that we can help make this study project a resounding success."M-WZ-003*

The participants expressed their acknowledgment and acceptance of the fact that the responsibility for the successful implementation of the TB elimination plan and practice lies primarily with them. They emphasized that this responsibility cannot be delegated or transferred to any third party. However, they also expressed deep appreciation for the support and contribution provided by partners in the ongoing fight against TB. The participants recognized the importance of collaborative efforts and acknowledged that the assistance from partners plays a significant role in achieving the goals of TB elimination.

*"… we believe that the valuable research conducted in this field, which plays a crucial role in enhancing our understanding and improving our strategies for TB prevention and treatment. Such support is highly valued and encourages us to further strengthen our efforts towards TB elimination"-RHB representative-M-WGZ-002.*

Thus, the optimization approach with this study believed to be an opportunity to strengthen the overwhelmed health system, logistics and supplies and enhance the capacity of health manpower. Accordingly, the stakeholders assured that they will own, lead, and jointly monitor the planned study to be successful.

**Considering cross-cutting multiple factors.**  According to the participants' reflection, the community in the study areas are highly supportive of the study aimed at TB elimination. However, it is important to acknowledge that solely introducing this intervention may not be sufficient due to various challenges faced by our people. These challenges include malnutrition, sociocultural factors, livelihood, and housing, low awareness of TB, and limited access to healthcare facilities for TB treatment. Therefore, it is crucial to adopt a comprehensive approach that addresses these underlying needs and engages the community effectively.

*"…I have a concern that our people are suffering from malnutrition, low awareness of the diseases itself and could not afford to come to the health facilities for TB treatment." One of the heads of zonal disease prevention and control." M-WZ-004*

On the other hand, one such factor is the existing inadequacy in healthcare workforce and logistics, which, if not well prepared for, can lead to an increased workload and frustration. Therefore, it is crucial to effectively manage these challenges by being conscious of these factors and taking appropriate measures to address them.

*"...It is crucial to consider the potential outcomes of community mobilization, including the potential for healthcare workers to face an increased workload, which may exacerbate their existing overwhelm. Additionally, there is a possibility of strain on testing kits and drugs due to the increased demand"* one of the health facility head. M-WGZ-003

Participants also emphasized the importance of recognizing that certain socio-economic realities in the community may persist despite efforts to demonstrate TB intervention or strengthen existing health programs.

*"For instance, the presence of congregated homes and communities, as well as health facilities that do not meet the required standards for providing quality health services for TB elimination, may pose challenges. While these factors may be beyond the immediate control of health programs, it is crucial to recognize and address them to effectively work towards the goal of TB elimination"* M-WGZ 004

Specifically, the participants emphasized the possible joint treatment of TB treatment and conditions of under nutrition. It was reiterated that TB is a disease of the poor where solution shoulder looked for the affected people with co-epidemic, TB, and malnutrition.

*''… now our farmers are deprived of food. However, this disease (means TB) could not be cured only with medication, but adequate food should be part of the TB treatment. Scarcity of food among our farmers is one of our headaches.'* M-WZ-004

Therefore, it is through considering these multiple factors that TB program implementers can ensure that their programs are comprehensive and effective and can more accurately reflect real-world situations and needs. It was learnt that TB cannot be eliminated only through TB program but the collaboration with other non- health sectors is of paramount importance.

**The importance of assigning TB treatment supporter.** According to the participants' reflection, one important aspect of successful TB treatment is the assignment of a treatment supporter. This is usually a family member or close friend who can help the person with TB adhere to their treatment plan. Treatment for TB typically involves taking multiple medications over a period of several months, and it can be challenging for someone to remember to take their medications every day. Treatment supporters can help ensure that the person with TB takes their medications as prescribed, which is crucial for the success of the treatment.

*"…Once patients are aware of the importance of taking the medication daily and the complication of TB drug interrupting, they will not do so. Hence, it is usually preferable to assign family members or a friend as treatment supporter or convince him to take the medications by himself."* One of the heads of zonal disease prevention and control." M-M-WGZ-005

In addition to helping with medication adherence, treatment supporters can also provide emotional support and encouragement to the person with TB. TB can be a stressful and isolating experience, and having someone there to offer support can make a significant difference.

*"The treatment supporters are not only for the sake of supervised treatment but also to deal with the possible stigma and discrimination due to TB diseases."* On of the community elders M-WAZ-006

## Discussion

This study reveals two broader categories. Firstly, the study reflects the insights gained from the MRC tailored protocol development, though the study team acknowledges the need for further data to fully support this observation. This includes the identification of effective strategies and approaches for addressing the research objectives. Secondly, the study emphasizes the importance of the MRC tailored feasibility and piloting, which provided valuable information on the practicality and viability of implementing the proposed interventions.

### Insights gained from the MRC tailored protocol development

The process of protocol development for the DeMONSTRATE TB study involved working with a variety of experts, researchers, the community, and regulatory bodies to ensure that the study plan is comprehensive and effective. This on the other hand found to be a complex and time-consuming process when it comes to applying the implementation research design in real-world settings [20], as it requires careful consideration of multiple perspectives and input from various stakeholders. Additionally, obtaining ethical clearance for the protocol from local and global IRBs was an overwhelming task, as it involves navigating complex regulations and ensuring that all ethical considerations are fully addressed. While the process of protocol development and ethical clearance was challenging due to the delay in review process, it is a crucial step in ensuring the efficacy and ethical integrity of any research project that involves human subjects.

Nevertheless, the lessons learnt in the process of several discussions during the development of this study protocol was the need of harmony among experts and decision makers from the beginning and that of in the process of protocol development to minimize unnecessary delay and without compromising the quality of research plan. However, the close follow-up and support from senior management at all levels were vital for the successful initiation of the study plan. In addition, early identification of the relevant study design would make the entire protocol development process to be simpler and smooth. Yet it is essential to have a 'tie breaker' in scenario of multiple experts' opinions and different approaches of executing research for a certain objective that considers the idea of the majority, as there could often be an outlier [21].

During protocol development, it became apparent and offered further insight for researchers to exercise caution in their planning by considering unforeseen needs that may necessitate the allocation of additional resources and time for navigating the ethical clearance process to ensure that the research is conducted with integrity and in compliance with established ethical standards.

Our results further validated existing difficulties with implementation of interventional studies such as financial inadequacy, multiple review cycles, obtaining IRB approvals, recruiting patients, complicated informed consent agreements, and completing substantial amounts of paperwork. It is crucial to acknowledge the evidence that these challenges not only have the potential to compromise the quality of the study plan but also to create frustration among investigative teams [22].

Thus, it is essential to consider unforeseen needs, such as allocating sufficient resources, securing funding, setting realistic timelines, strengthening communication and collaboration, holding regular progress meetings, and implementing proactive risk management strategies.

These measures are crucial for promptly identifying and addressing issues, as well as mitigating potential challenges. Fostering a supportive team culture is also crucial for maintaining the quality of the study plan. Overall, there is a need for preparedness, perseverance and determination, these barriers can be overcome to advance medical research and improve patient outcomes. Also, there should be agreed upon scheduled milestones that the study team should stick to.

## Practicality and viability of implementing the proposed interventions

**Government interest toward local evidence and community engagement.** During the piloting phase, the stakeholders were of the belief that "looking for local evidence through participation of local people is compulsory for TB elimination through". This highlights the recognition that community engagement plays a crucial role in achieving success in TB elimination efforts. Studies suggest that involving influential community members during the early stages of planning of a study in community-based TB clinical trials can improve the acceptability and relevance of TB research [23]. Overall, the innovative community-based interventions with community engagement have been shown to have a positive impact on the TB diagnostic and treatment services with dramatical increment in the TB case notification rate and improving treatment outcome [24,25]. When the community is engaged, it can help to identify and address barriers to access, such as transportation or social stigma [26].

**Considering cross-cutting multiple factors.** The participants have also recognized that the government's interest in local evidence is essential to understanding the unique factors that contribute to the spread of TB in different settings. To make progress towards TB elimination study through the demonstration of multi-pronged interventions, officials must actively engage the community and consider a variety of factors that influence overall TB outcomes. These factors include nutrition, sociocultural factors, livelihood, and housing and these are also identified in other studies [2,19,24]. By considering these factors and working closely with the community, stakeholders believe that it is possible to make tangible progress in the fight against TB.

**Homogeneity of officials' commitment across the regions.** One important aspect emphasized in this study is the need for government officials at all levels to show a commitment to eliminating TB. Having a consistent approach and unwavering commitment across diverse regions is key to ensuring that efforts are coordinated, effective, and efficient in resource utilization. According to the research conducted in Ghana [27], and end TB strategy of India [28] it was found that the involvement and dedication of political authorities in advocacy and resource allocation have played a significant role in the effective control of TB. That is, the engagement of decision makers can create ownership and assist in the sustainability of the implementation of innovative TB prevention and control.

These studies demonstrated that when political leaders actively participate and show commitment, it positively influences the success of TB control efforts in the country. This highlights the importance of having political support and engagement in tackling the challenges posed by TB. This later could simplify the acceptance of policy briefs and frameworks extracted from this study to be endorsed by the national TB program.

**The importance of assigning TB treatment supporter.** Establishing treatment supporters is an important aspect of TB elimination programs because it helps patients to complete their treatment successfully and improves their treatment outcomes. These supporters can take many forms, such as family members, friends, healthcare professionals, or community health workers based on the patient's own inclination of deciding whom to be a treatment supporter [29]. By providing support and encouragement throughout the treatment process, supporters can help patients to stay motivated and committed to completing their medication

regimen. They can also help patients to manage any side effects or other issues that may arise during treatment including addressing physical, emotional, and social needs. By improving treatment outcomes in this way, TB elimination programs can have a real impact on public health, reducing the spread of TB and improving the overall health of affected communities. Recent studies have shown that having a treatment supporter can significantly improve the outcomes of patients with TB and or MDR TB [30]. Hence, treatment supporters could also be considered for TPT at the community level.

In conclusion, our findings highlight several key considerations for designing multi-pronged TB elimination interventions. Firstly, addressing varying levels of knowledge and experience among experts through enhanced communication mechanisms is crucial. Regular meetings and group discussions can ensure alignment and foster open dialogue. Secondly, the stepwise approach of the MRC framework was essential in ensuring study objectives, methods and milestones are adjusted prior to the main study phase Thirdly, involving all stakeholders in decision-making when changes to the project design occur, while considering the majority opinion, helps maintain project focus and minimize conflicts. Community engagement and dedicated officials are crucial in effectively addressing TB in implementation research. Considering factors like nutrition, sociocultural influences, livelihood, and housing is vital to address the complexity of TB and improve treatment and prevention efforts. Therefore, it is recommended that community engagement, commitment from TB program managers, and the implementation of multifaceted interventions require even greater attention during the adoption and implementation of the TB elimination framework.

## Supporting information

**S1 Table. Profile of study districts.** S1 Table indicates the coordinates of the study districts to prepare their maps, their population, health facilities and TB burden.
(DOCX)

**S2 Table. Question guide template for the experience of MRC tailored feasibility and piloting.** S2 Table indicates the template used by the researcher as a question guide and compiled the interview opinions and information. While brainstorming, detailed notes were captured in the transcription column. There is a discussion row where question guides were set. After the end of the discussion, the researchers noted down the code in the code column and summarized categories in the theme column.
(DOCX)

**S3 Table. List of participants during the participatory discussion.** Their names are coded, and the phone numbers are removed. In the excel, the sheet with 'List_Boloso Bombe (BB)' refers to 'S3 Table BB. List of participants during the participatory discussion Boloso Bombe Study'; sheet with 'List_Gedeb Hassassa (GH)' refers to 'S3 Table HH. List of participants during the participatory discussion Gedeb Hasasa woreda DeMONSTRATE TB Study Launching'; and the sheet with 'List_ Jabi Tihnan (JT)' refers to 'S3 Table JT. List of participants during the participatory discussion: Jabi Tihnan Woreda/Finote Selam Study Launching'.
(XLSX)

## Acknowledgments

The authors wish to thank the Oromia, Amhara, Southern Ethiopia regions' health bureaus, zonal offices, woreda health desk, and health posts for their strong coordination with the USAID Eliminate TB project. The authors are so grateful for the participants of this study.

This study is made possible by the generous support of the American people through the United States Agency for International Development (USAID) under cooperative agreement #72066320CA00009. The contents of this study are the responsibility of Management Sciences for Health (MSH) and do not necessarily reflect the views of USAID or the United States Government.

## Author contributions

**Conceptualization:** Daniel G. Datiko, Degu Jerene, Wondmiu Gebrekiros, Anteneh Kassa, Yewulsew Kassie.

**Data curation:** Mulatu Biru, Zewdu G. Dememew.

**Formal analysis:** Mulatu Biru, Zewdu G. Dememew.

**Funding acquisition:** Daniel G. Datiko, Wondmiu Gebrekiros.

**Investigation:** Mulatu Biru, Degu Jerene, Asfawesen G. Yohannes, Yohannes Molla, Pedro Suarez.

**Methodology:** Mulatu Biru, Daniel G. Datiko, Degu Jerene, Asfawesen G. Yohannes, Wondmiu Gebrekiros, Anteneh Kassa, Yewulsew Kassie, Zewdu G. Dememew.

**Project administration:** Mulatu Biru, Asfawesen G. Yohannes, Pedro Suarez, Zewdu G. Dememew.

**Resources:** Daniel G. Datiko, Yewulsew Kassie.

**Supervision:** Mulatu Biru, Daniel G. Datiko, Degu Jerene, Asfawesen G. Yohannes, Yohannes Molla, Pedro Suarez, Zewdu G. Dememew.

**Validation:** Mulatu Biru, Degu Jerene, Yohannes Molla, Pedro Suarez.

**Visualization:** Degu Jerene, Yohannes Molla, Pedro Suarez.

**Writing – original draft:** Mulatu Biru, Daniel G. Datiko, Degu Jerene, Zewdu G. Dememew.

**Writing – review & editing:** Mulatu Biru, Daniel G. Datiko, Degu Jerene, Asfawesen G. Yohannes, Zewdu G. Dememew.

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
