## [Decision Letter · Decision Letter 0]

29 Nov 2024

PONE-D-24-24730Designing Tuberculosis elimination framework through participatory processes in Ethiopia: results from stakeholders’ discussionsPLOS ONE

Dear Dr. Dememew,

Thank you for submitting your manuscript to PLOS ONE. After careful consideration, we feel that it has merit but does not fully meet PLOS ONE’s publication criteria as it currently stands. Therefore, we invite you to submit a revised version of the manuscript that addresses the points raised during the review process.

Please submit your revised manuscript by Jan 13 2025 11:59PM. If you will need more time than this to complete your revisions, please reply to this message or contact the journal office at plosone@plos.org . Please include the following items when submitting your revised manuscript:

We look forward to receiving your revised manuscript.

Kind regards,

Mohammed Hasen Badeso, Epidemiologist

Academic Editor

PLOS ONE

2. In the ethics statement in the Methods, you have specified that verbal consent was obtained. Please provide additional details regarding how this consent was documented and witnessed, and state whether this was approved by the IRB.

3. Please amend the manuscript submission data (via Edit Submission) to include authors Asfawesen G.Yohannes, Wondmiu G. Kirstos, Anteneh K, and Yewulsew K.

4. Please amend your authorship list in your manuscript file to include author Degu Gilmore Jeren.

5. We note that Figure 2 in your submission contain [map/satellite] images which may be copyrighted. All PLOS content is published under the Creative Commons Attribution License (CC BY 4.0), which means that the manuscript, images, and Supporting Information files will be freely available online, and any third party is permitted to access, download, copy, distribute, and use these materials in any way, even commercially, with proper attribution. For these reasons, we cannot publish previously copyrighted maps or satellite images created using proprietary data, such as Google software (Google Maps, Street View, and Earth). For more information, see our copyright guidelines: http://journals.plos.org/plosone/s/licenses-and-copyright.

1. You may seek permission from the original copyright holder of Figure 2 to publish the content specifically under the CC BY 4.0 license. 

Reviewers' comments:

Reviewer's Responses to Questions

**Comments to the Author**

1. Is the manuscript technically sound, and do the data support the conclusions?

Reviewer #1: Yes

Reviewer #2: No

2. Has the statistical analysis been performed appropriately and rigorously? 

Reviewer #1: N/A

Reviewer #2: Yes

3. Have the authors made all data underlying the findings in their manuscript fully available?

Reviewer #1: Yes

Reviewer #2: No

4. Is the manuscript presented in an intelligible fashion and written in standard English?

Reviewer #1: Yes

Reviewer #2: No

5. Review Comments to the Author

Reviewer #1: Date 20 September 2024

PLOS ONE Journal

Dear Editors

Thank you so much for giving me the opportunity to review the manuscript in title “Designing Tuberculosis elimination framework through participatory processes in Ethiopia: results from stakeholders’ discussions.”

The study has strength and limitations.

strength

Background : defined the problem and principles

Methods: presented detailed information on data acquisition and analysis

Results: Themes and subthemes with summary of finding and verbatims depicted.

Discussion: findings were controlled with relevant literatures.

Conclusions: proposed in line with results.

Limitation

Methods: statement describing quasi- experimental nature of this study should be removed. Here is no intervention and control groups.

Data saturation: the researcher dealt with three groups in three region, statement stating data saturation cannot be reached with a single group.

Ethical Consideration: the annexed certificate was issued by PSI not by JSI. But in the body of the manuscript you reported that John Snow Inc and AHRI ethics review boards.

Thank you so much.

Reviewer #2: Introduction

• Global TB report data pertains to 2022, so the statement may be modified accordingly

• Many of the determinants mentioned are not primarily related to treatment outcomes, but to TB incidence. The interventions for improving treatment outcomes may be kept separate from all the determinants of TB

Results

• “five main themes” in table 3 mentioned in the manuscript is mentioned in the previous paragraph as “sub-themes”

• “approval from regional research approving bodies in addition to the national IRB, can be stopped” Obtaining clearance from regional and national IRBs can’t be equated to community involvement

Discussion

• “The study team learnt that engaging health care workers from the study district and the community is cost effective and eases the implementation of the study” – based on what evidence?

• Results do not have data to substantiate the statement that “study highlights the valuable insights gained from the MRC tailored protocol development.”

• Again the results and the quotes presented in the results do not substantiate the statement that “the study emphasizes the significance of the MRC tailored feasibility and piloting, which provided valuable information on the practicality and viability of implementing the proposed interventions”

Abstract

• Abstract does not convey what the study involves and the conclusions overreach the findings of the study

6. PLOS authors have the option to publish the peer review history of their article (what does this mean? ). If published, this will include your full peer review and any attached files.

**Do you want your identity to be public for this peer review?** For information about this choice, including consent withdrawal, please see our Privacy Policy .

Reviewer #1: **Yes: ** Dr Mesele Damte Argaw

Reviewer #2: No

---

## [Author Response · Author response to Decision Letter 0]

8 Jan 2025

A rebuttal letter

Thank you for the opportunity to respond to the editor and reviewers' comments on our manuscript titled " Designing Tuberculosis elimination framework through participatory processes in Ethiopia: results from stakeholders’ discussions” Manuscript ID: [PONE-D-24-24730].

We appreciate the reviewers’ constructive feedback, which has helped us to improve the clarity and quality of our work. Below, we provide a detailed response to each comment. We have carefully considered all suggestions and revised the manuscript accordingly. For clarity, the editor and reviewers' comments are presented in bold italics, followed by our point-by-point responses.

A. Response to Editor:

Response: Thank you for pointing this out. We have carefully reviewed and revised our manuscript to ensure full compliance with PLOS ONE's style requirements, including the appropriate file naming conventions.

2. In the ethics statement in the Methods, you have specified that verbal consent was obtained. Please provide additional details regarding how this consent was documented and witnessed, and state whether this was approved by the IRB.

Response: Thank you for your comment. Verbal consent was obtained from all participants prior to data collection. The consent process was documented by the research team through written records indicating the participant's agreement, which weas signed by both the interviewer and a witness during the consent process. This procedure was part of the study Demonstrating Multipronged and Optimized Novel Strategies to Reinforce Actions Targeted at Eliminating Tuberculosis (DeMONSTRATE-TB): A Sequential Exploratory Mixed Method Study in Ethiopia (REB# 37.2022) and was reviewed and approved through the human subjects expedited review process on December 16, 2022, and modification request approval in May 31, 2023.

3. Please amend the manuscript submission data (via Edit Submission) to include authors Asfawesen G.Yohannes, Wondmiu G. Kirstos, Anteneh K, and Yewulsew K.

Response: Thank you for your comment. The manuscript submission data has been updated to include authors Asfawesen G. Yohannes, Wondmiu G. Kirstos, Anteneh K., and Yewulsew K.

4. Please amend your authorship list in your manuscript file to include author Degu Gilmore Jeren.

Response: The author's name has been amended to Degu Jerene.

5. We note that Figure 2 in your submission contain [map/satellite] images which may be copyrighted. All PLOS content is published under the Creative Commons Attribution License (CC BY 4.0), which means that the manuscript, images, and Supporting Information files will be freely available online, and any third party is permitted to access, download, copy, distribute, and use these materials in any way, even commercially, with proper attribution. For these reasons, we cannot publish previously copyrighted maps or satellite images created using proprietary data, such as Google software (Google Maps, Street View, and Earth). For more information, see our copyright guidelines: http://journals.plos.org/plosone/s/licenses-and-copyright.

We require you to either (1) present written permission from the copyright holder to publish these figures specifically under the CC BY 4.0 license, or (2) remove the figures from your submission

Response: Thank you for your comment. This figure was created by us using QGIS (https://qgis.org/), which is open-source software. Please note that the boundaries in this map do not represent official boundaries. We have also included this information in the figure caption

6. Please include captions for your Supporting Information files at the end of your manuscript, and update any in-text citations to match accordingly.

Response: Thank you for this information. We do not have any additional supporting documents for Figure 2.

7. Please review your reference list to ensure that it is complete and correct. If you have cited papers that have been retracted, please include the rationale for doing so in the manuscript text or remove these references and replace them with relevant current references. Any changes to the reference list should be mentioned in the rebuttal letter that accompanies your revised manuscript. If you need to cite a retracted article, indicate the article’s retracted status in the References list and include a citation and full reference for the retraction notice.

Response: Thank you for your comment. We have reviewed the reference list to ensure completeness and accuracy. No retracted papers were cited. Please let us know if further adjustments are required

B. Response From Reviewers

Response to Reviewer 1:

Thank you so much for giving me the opportunity to review the manuscript in title “Designing Tuberculosis elimination framework through participatory processes in Ethiopia: results from stakeholders’ discussions.”

The study has strength and limitations.

1.0 strength

Background : defined the problem and principles

Methods: presented detailed information on data acquisition and analysis

Results: Themes and subthemes with summary of finding and verbatims depicted.

Discussion: findings were controlled with relevant literatures.

Conclusions: proposed in line with results.

Response: We thank you the reviewer for the constructive comments

2.0 Limitation

Methods: statement describing quasi- experimental nature of this study should be removed. Here is no intervention and control groups.

Response: Thank you removed, line 135

Data saturation: the researcher dealt with three groups in three regions, statement stating data saturation cannot be reached with a single group.

Response: Thank you. This saturation is for context specific or districts-based information. However, for cross-cutting information, discussion in all the districts were used-line 200-201.

Ethical Consideration: the annexed certificate was issued by PSI not by JSI. But in the body of the manuscript, you reported that John Snow Inc and AHRI ethics review boards.

Response: Thank you it is corrected to Population Services International (PSI), line 222-223

Response to Reviewer 2:

Introduction

1.0 Global TB report data pertain to 2022, so the statement may be modified accordingly

Response: Corrected in Line 66

2.0 Many of the determinants mentioned are not primarily related to treatment outcomes, but to TB incidence. The interventions for improving treatment outcomes may be kept separate from all the determinants of TB

Response: This is acritical comment and addressed in line #67-77.

Results

1.0 “five main themes” in table 3 mentioned in the manuscript is mentioned in the previous paragraph as “sub-themes”

Response: Thank you, and this is corrected as 'five sub-themes' in figure 3. Line# 296

2.0 “approval from regional research approving bodies in addition to the national IRB, can be stopped” Obtaining clearance from regional and national IRBs can’t be equated to community involvement

Response: Thank you very much. New supporting quates added to deal with the importance of community engagement. Line 347-350

Discussion

1. The study team learnt that engaging health care workers from the study district and the community is cost effective and eases the implementation of the study” – based on what evidence?

Response: Thank you for this critical observation. We have revised the statement in line #353 to clarify that the study team learned that engaging healthcare workers from the study district and the community facilitated and streamlined the implementation of the study.

2. Results do not have data to substantiate the statement that “study highlights the valuable insights gained from the MRC tailored protocol development.”

Response: Thank you again for this crucial comment regarding the connection between evidence and data. We have revised the statement in line #478-480 to: "The study reflects the insights gained from the MRC tailored protocol development, though we acknowledge the need for further data to fully support this observation."

3. Again, the results and the quotes presented in the results do not substantiate the statement that “the study emphasizes the significance of the MRC tailored feasibility and piloting, which provided valuable information on the practicality and viability of implementing the proposed interventions.

Response: We greatly appreciate this comment and have made a minor change, replacing "significance" with "importance" in line # 483. However, based on the evidence presented in the results—such as the MRC tailored protocol development process, which enhanced the rigor of the protocol, and the feasibility discussions—it is evident that community engagement at various levels played a crucial role in facilitating the commencement of the study.

4. Abstract does not convey what the study involves, and the conclusions overreach the findings of the study

Response: Thank you once again. The conclusion of the abstract has been revised to encompass the key findings of the study, as reflected in lines #49-54.

---

## [Editor Report · Decision Letter 1]

12 Jan 2025

Designing Tuberculosis elimination framework through participatory processes in Ethiopia: results from stakeholders’ discussions

PONE-D-24-24730R1

Dear Dr. Zewdu

We’re pleased to inform you that your manuscript has been judged scientifically suitable for publication and will be formally accepted for publication once it meets all outstanding technical requirements.

Kind regards,

Mohammed Hasen Badeso, Epidemiologist

Academic Editor

PLOS ONE
---

## [Editor Report · Acceptance letter]

PONE-D-24-24730R1

PLOS ONE

Dear Dr. Dememew,

I'm pleased to inform you that your manuscript has been deemed suitable for publication in PLOS ONE. Congratulations! Your manuscript is now being handed over to our production team.

Kind regards,

on behalf of

Mr Mohammed Hasen Badeso

Academic Editor

PLOS ONE